# US primary care in 2029: A Delphi survey on the impact of machine learning

**Charlotte Blease**[1]*, **Anna Kharko**[2], **Cosima Locher**[3,4], **Catherine M. DesRoches**[1,5], **Kenneth D. Mandl**[6,7,8]

1 Division of General Medicine, OpenNotes, Beth Israel Deaconess Medical Center, Boston, MA, United States of America, 2 Department of Psychology, University of Plymouth, Plymouth, United Kingdom, 3 Division of Clinical Psychology and Psychotherapy, University of Basel, Basel, Switzerland, 4 Department of Anesthesiology, Boston Children's Hospital, Boston, MA, United States of America, 5 Harvard Medical School, Boston, MA, United States of America, 6 Department of Biomedical Informatics, Harvard Medical School, Boston, MA, United States of America, 7 Department of Pediatrics, Harvard Medical School, Boston, MA, United States of America, 8 Computational Health Informatics Program, Boston Children's Hospital, Boston, MA, United States of America

* cblease@bimdc.harvard.edu

**Data Availability Statement:** Individual participant data for Rounds 1 and 3 of the Delphi Poll cannot be shared publicly because of the need to ensure participant anonymity (per Beth Israel Deaconess Medical Center Institutional Review Board). De-identified raw data from Round 2 is available as S3

## Abstract

### Objective

To solicit leading health informaticians' predictions about the impact of AI/ML on primary care in the US in 2029.

### Design

A three-round online modified Delphi poll.

### Participants

Twenty-nine leading health informaticians.

### Methods

In September 2019, health informatics experts were selected by the research team, and invited to participate the Delphi poll. Participation in each round was anonymous, and panelists were given between 4–8 weeks to respond to each round. In Round 1 open-ended questions solicited forecasts on the impact of AI/ML on: (1) patient care, (2) access to care, (3) the primary care workforce, (4) technological breakthroughs, and (5) the long-future for primary care physicians. Responses were coded to produce itemized statements. In Round 2, participants were invited to rate their agreement with each item along 7-point Likert scales. Responses were analyzed for consensus which was set at a predetermined interquartile range of ≤ 1. In Round 3 items that did not reach consensus were redistributed.

### Results

A total of 16 experts participated in Round 1 (16/29, 55%). Of these experts 13/16 (response rate, 81%), and 13/13 (response rate, 100%), responded to Rounds 2 and 3, respectively. As a result of developments in AI/ML by 2029 experts anticipated workplace changes

Appendix. All relevant aggregated data are within the manuscript and its Supporting Information files.

**Funding:** CB was supported by an Irish Research Council-Marie Skłodowska-Curie Fellowship, and a Keane Scholar Award. AK was funded by the School of Psychology, University of Plymouth. CL was funded by a Swiss National Science Foundation grant (P400PS_180730). The study funders played no role in the study design; writing of the report; or the decision to submit the manuscript for publication. Researchers were independent of influence from study funders.

**Competing interests:** The authors have declared that no competing interests exist.

including incursions into the disintermediation of physician expertise, and increased AI/ML training requirements for medical students. Informaticians also forecast that by 2029 AI/ML will increase diagnostic accuracy especially among those with limited access to experts, minorities and those with rare diseases. Expert panelists also predicted that AI/ML-tools would improve access to expert doctor knowledge.

## Conclusions

This study presents timely information on informaticians' consensus views about the impact of AI/ML on US primary care in 2029. Preparation for the near-future of primary care will require improved levels of digital health literacy among patients and physicians.

## Introduction

### Background

Attention in medicine and related fields has increasingly focused on the potential of big data, artificial intelligence (AI), and machine learning (ML) to change the delivery of healthcare [1–4]. Much of this debate has focused on the promise of AI/ML to augment or even disintermediate the clinical roles of physicians in gathering and monitoring patient health information, and to undertake core tasks such as diagnostics, prognostics, and the formulation of personalized patient healthcare plans [1, 4–9]. Differentiating the hype from hope in the discourse about AI/ML in medicine is crucial to better understand the scope for the computerization of medicine. Although broad predictions of the impact of AI/ML on healthcare are ubiquitous, credible short-term predictions are necessary to address questions about resource allocation, and the adequacy of medical education and training.

### Objectives

Recently a number of surveys have explored medical students', and physicians' views about the impact of AI/ML on the future of medical practice [10–16]. Currently, there is scarce exploration of consensus views among informaticians [17]; in particular, on how AI/ML might meaningfully influence medical care in the short-term [18]. To address this research gap, we designed a Delphi survey to explore leading health informaticians' predictions about the impact of machine learning on primary care in the US in 2029. To our knowledge, this is the first investigation of experts' opinions about the impact of AI/ML on the future of the near future of general medical practice.

### Methods

#### The Delphi method

The Delphi Method, developed by the Rand Corporation in the 1950s [19, 20] is designed to pool the opinions of a purposive sample of identified experts in a given field to establish consensus predictions [21, 22]. Delphi polls rely on non-probability sampling techniques to identify a panel of experts: since participants are not randomly selected, representativeness is neither intended nor assured [23]. The selected panel of experts is invited to answer a series of questions anonymously [23]. Participants are next asked to reassess their initial judgments in light of group trends until consensus is obtained [24]. This anonymous, iterative technique

carries distinctive advantages over focus groups by avoiding the influences of individual dominant personalities, group-think, and helps to keep participants 'on topic' [19, 23].

Delphi surveys are particularly well suited to exploring consensus views related to new lines of inquiry, and for establishing goal-setting, and needs assessments in policy-making [19, 23]. Since Delphi polls provide more accurate predictions than other forecasting methodologies, the approach is often used as a policy and practice heuristic for health care management, and resource allocation [25–27].

## Approach

We used a modified Delphi technique which is structured into three discrete rounds [19, 23, 28, 29]. In Round One, questions are open-ended, requiring free-text answers. Responses are aggregated and coded into a series of statements. In Round Two, experts are provided with this list of statements, and requested to provide their level of agreement with each item. Depending on the survey items, Round 2 and 3 questionnaires requested 'yes' or 'no' responses, or participants' level of agreement with statements on 7-point Likert scales: 1 = *greatly decrease*, 2 = *moderately decrease*, 3 = *slightly decrease*, 4 = *remain the same*, 5 = *slightly increase*, 6 = *moderately increase*, 7 = *greatly increase*; 1 = *very unlikely*, 2 = *moderately unlikely*, 3 = *slightly unlikely*, 4 = *uncertain*, 5 = *slightly likely*, 6 = *moderately likely*, 7 = *very likely*; or 1 = *strongly disagree*, 2 = *moderately disagree*, 3 = *slightly disagree*, 4 = *neutral*, 5 = *slightly agree*, 6 = *moderately agree*, 7 = *strongly increase*. Those statements that reach a predefined level of agreement are omitted to reduce participant survey fatigue, and items that lack consensus are re-circulated via a final anonymous poll. In the third and final round, panelists are reminded of their own response to the remaining statements as well as the median response of other experts, and are invited to preserve or revise their answer. A key aim of Delphi methodology is to maintain as high a response rate as possible rounds [23, 30, 31], and the accuracy of forecasts has been demonstrated to improve between each round [32].

## The expert panel

Although there is no universally agreed sample size for Delphi polls [23], our aim was to balance the size of the panel with a high response rate between the three rounds. We therefore aimed to achieve a panel of around 12–15 individuals who would agree to share their expertise, and be committed to giving their time to respond to each round. Using purposive sampling methodology, the research team compiled a list of 27 highly trained and knowledgeable individuals with context-specific knowledge about health informatics and primary care in the US. Addressing the question about how to identify domain-specific 'experts', our goal was to prioritize panelists for their recognized competence in the field of health informatics. We defined expertise to mean a person who had published significant contributions within the field of health informatics, and/or individuals who were currently appointed as research leaders, or as health information officers. Acknowledging that heterogeneous panels have been shown to result in more accurate estimates [33], and that what counts as an expert can be influenced by goals, values, and the manner in which knowledge is generated, we aimed to recruit diverse participants from across academia, healthcare, non-profit organizations, and industry; and to strive for panelists with a varied complementarity of interests within health informatics. Measures were also taken to ensure demographic diversity among invited participants along the lines of gender, age, nationality, and race/ethnicity. This study was deemed exempt research by Beth Israel Deaconess Medical Center Institutional Review Board and granted ethical approval by the University of Plymouth, UK.

Prospective panelists were contacted via email in September 2019, with an invitation and internet link to the survey. Individuals were informed that we desired a commitment on the part of experts to respond to all three rounds, that adequate response time would be given to answer each round of the survey, participation was voluntary and unpaid, and that participants could withdraw at any time. Prospective participants were also informed that they would remain anonymous to other participants, their individual responses would not be shared with other panelists, and their contribution would be confidential. Respondents' names were also replaced with a study ID number by AK in order to preserve participant anonymity among other team members in data analysis.

## The questionnaire

We created an electronic questionnaire on JISC Online Surveys hosted at the University of Plymouth, UK (https://www.onlinesurveys.ac.uk/). The poll incorporated a three-step modified Delphi method which took place between September 2019 and January 2020. Participants were sent 3 reminders after each round of the survey, and given 4–6 weeks to respond to Rounds 1 and 2, and 8 weeks to respond to Round 3 which fell over the New Year period.

In the first round, the Delphi survey requested demographic information; this was followed by 5 sections, with 7 open-ended questions, on the impact of machine learning on primary care by 2029 (see S1 Appendix; Table 1). The sections comprised: (1) Patient care (3 open-ended questions); (2) Access to care (1 open-ended question); (3) Primary care workforce (1 open-ended question); (4) Technological breakthroughs (1 open-ended question); and (5) The future of the primary care physicians (1 closed ended question; and 1 open-ended question). We also included a final comment-box for feedback on the survey.

**Table 1. Round 1 questions.**

| Item |
|---|
| **Patient Care** |
| *By 2029, in your opinion, please predict the effect(s)–if any–of machine learning/AI on diagnostic accuracy in primary care in the USA. Please describe 1 or 2 predictions, briefly, below.* |
| *By 2029, in your opinion, please predict the effect(s)–if any–of machine learning/AI on health care disparities in the USA. Please describe 1 or 2 predictions, briefly, below.* |
| *By 2029, in your opinion, please predict the effects–if any–of machine learning/AI on the empathic care of primary care patients in the USA. Please describe 1 or 2 predictions, briefly, below.* |
| **Access to Care** |
| *By 2029, in your opinion, please predict the effects—if any—of machine learning/AI on patient access to medical care in the USA. Please describe 1 or 2 predictions, briefly, below.* |
| **Primary Care Workforce** |
| *By 2029, in your opinion, please predict the effects–if any–of machine learning/AI on the composition of the primary care workforce in the USA. Please describe 1 or 2 predictions, briefly, below.* |
| **Technological Advancements in Primary Care** |
| *In your opinion, please predict what—if any—major AI breakthroughs would be important to improve diagnostic accuracy in medicine? Please provide at least one or two important breakthroughs, or if you believe no such breakthroughs are necessary, please elaborate.* |
| **The Long-term Future of the Profession** |
| *In your opinion, will primary care doctors ever become obsolete?* |
| *If you answered 'yes', please give your best forecast of how many years from now this might happen. Please also provide a brief reason for your estimate.* |
| *If you answered 'no', please elaborate.* |
| *If you answered 'don't know', please elaborate.* |

Responses to Round 1 were collated and coded into lists of statements. Coding was conducted by CB and independently reviewed by CL and AK, and subsequent revisions were made. Comments that were unrelated to the themes, or were deemed redundant were eliminated. Similar statements were grouped together and translated into concise items; whenever possible, replication of exact phrasing by participants was employed. These items were circulated in Round 2, and an online survey was sent to each individual member of the panel. Participants were requested to respond to categorical variables by selecting a 'yes' or 'no' response, and to questions with continuous variables by using predefined 7-point Likert scales (see S2 Appendix).

Prior to consensus analysis of responses in Rounds 2 and 3, for categorical variables consensus was set at $\geq$ 75%, and for continuous variables consensus along 7-point semantic differential scales was set at an interquartile range of $\leq$ 1 [19, 34]. After analysis of Round 2 results, items that did not reach consensus were redistributed for Round 3. In Round 3, each participant received a personalized survey link. Panelists were reminded of their response to items in Round 2, and provided with the median collated response of the other participants.

## Results

### Round 1

We obtained 17/29 response for Round 1 (see Table 2). One invited expert circulated the online survey to another respondent who was later excluded bringing the total to 16/29 (55%) (see Fig 1). Round 1 comprised of 4 (25%) female and 12 (75%) male participants. Respondents differed from invited non-respondents in terms of gender: initial invitations were extended to 11 (38%) females and 18 (62%) male experts. All 16 panelists in Round 1 held an MD (n = 11, 69%), PhD (n = 9, 56%), or both (n = 1, 6%) (also see: Acknowledgments).

Responses to Round 1 were translated into itemized lists of statements. As a result of this process, the was survey was expanded into 57 items arranged into 7 sections: (i) diagnostic accuracy (10 items), (ii) healthcare disparities (5 items), (iii) empathic care of patients (8 items), (iv) access to care (9 items), (v) primary care workforce (7 items), (vi) technological advancements in primary care (10 items), and (vii) the long-term future of the profession (8 items) (see S2 Appendix). The panel was repeatedly prompted to forecast changes to primary care by 2029, and questions emphasized that predictions should be restricted to the US context. Throughout the survey experts were reminded to "predict what you believe **will happen** and ***not what you personally*** would like to see happen". After completing each section, participants were also invited to provide free text comments, and following completion of the survey, offered to provide any additional feedback.

### Rounds 2 and 3

In Round 2, 13/16 experts participated in the online survey (response rate of 81%) [see S3 Appendix, for Round Two raw data]. In Round 3, 13/13 experts responded (response rate of 100%). In Rounds 2 and 3, participants included 4 (31%) females, and 9 (69%) male participants (see Table 2 for demographic information). Table 3 presents the item means and standard deviations for item responses, and also indicates the items that reached consensus in Round 2, those that obtained consensus in Round 3, and items that failed to secure expert consensus. As described in the Methods, and as indicated in Table 3, items reflect three different 7-point Likert Scales. To undertake interpretation of panelists' predictions, these items were divided into three rational, a priori categories. For the scale identified as 'I', responses were bounded into items that experts expected to *increase* (item mean of 4.5 and greater), *remain about the same* (item mean of 3.5–4.4), and to *decrease* (item mean of 3.4 and less). For the

**Table 2. Demographic information.**

| | Round 1 (*n* = 16) [1] | | Rounds 2 & 3 (*n* = 13) | |
|---|---|---|---|---|
| | *n* or *m* | (% or *SD*) | *n* or *m* | (% or *SD*) |
| *Gender (n male)* | 12 | (75%) | 9 | (69%) |
| *Age (m years)* | 49.06 | (10.52) | 48.92 | (11.08) |
| *Nationality* [2] | | | | |
| Australia | 1 | (6%) | - | - |
| Switzerland | 1 | (6%) | 1 | (8%) |
| Taiwan | 1 | (6%) | 1 | (8%) |
| UK | 1 | (6%) | 1 | (8%) |
| USA | 13 | (81%) | 11 | (85%) |
| *Ethnicity* | | | | |
| Asian | 2 | (13%) | 2 | (15%) |
| Hispanic | 1 | (6%) | - | - |
| Mixed | 1 | (13%) | - | - |
| White | 12 | (69%) | 11 | (85%) |
| *MD Degree (total n)* [2] | 10 | (63%) | 7 | (54%) |
| Clinical Informatics | 2 | (13%) | 1 | (8%) |
| Emergency Medicine | 1 | (6%) | 1 | (8%) |
| General / Internal Medicine | 4 | (25%) | 3 | (23%) |
| Oncology | 1 | (6%) | 1 | (8%) |
| Pathology | 1 | (6%) | 1 | (8%) |
| Pediatrics | 1 | (6%) | 1 | (8%) |
| Surgery | 1 | (6%) | - | - |
| Unspecified | 1 | (6%) | - | - |
| *PhD Degree (total n)* [2] | 10 | (63%) | 9 | (69%) |
| Chemistry | 1 | (6%) | 1 | (8%) |
| Computer Science / Informatics | 6 | (38%) | 5 | (39%) |
| Linguistics / Cognitive Science | 1 | (6%) | 1 | (8%) |
| Medicine / Public Health | 1 | (6%) | 1 | (8%) |
| Physics | 1 | (6%) | 1 | (8%) |
| *Current job field* [2] | | | | |
| Academia | 13 | (81%) | 10 | (77%) |
| Government | 1 | (6%) | 1 | (8%) |
| Medicine | 2 | (13%) | 1 | (8%) |
| Non-profit Organisation | 1 | (6%) | 1 | (8%) |

n–count, m–average value per sample, %—percentage of the sample, rounded to the nearest whole value, SD–standard deviation. Average value and SD were calculated only for age.

[1] Round 1 calculations exclude the one non-eligible respondent.

[2] Questions for which some participants selected more than one option.

scale identified as 'L', responses were differentiated into items that the experts predicted to be *likely* (item mean of 4.5 and greater), items that they were *uncertain* about (item mean of 3.5–4.4), and those they predicted to be *unlikely* (item mean of 3.4 and less). Finally, for the scale identified as 'A', responses were bounded into items about which the panel *agreed* (item mean of 4.5 and greater), those that they were *neutral* on (item mean of 3.5–4.4), and those items about which they *disagreed* (item mean of 3.4 and less).

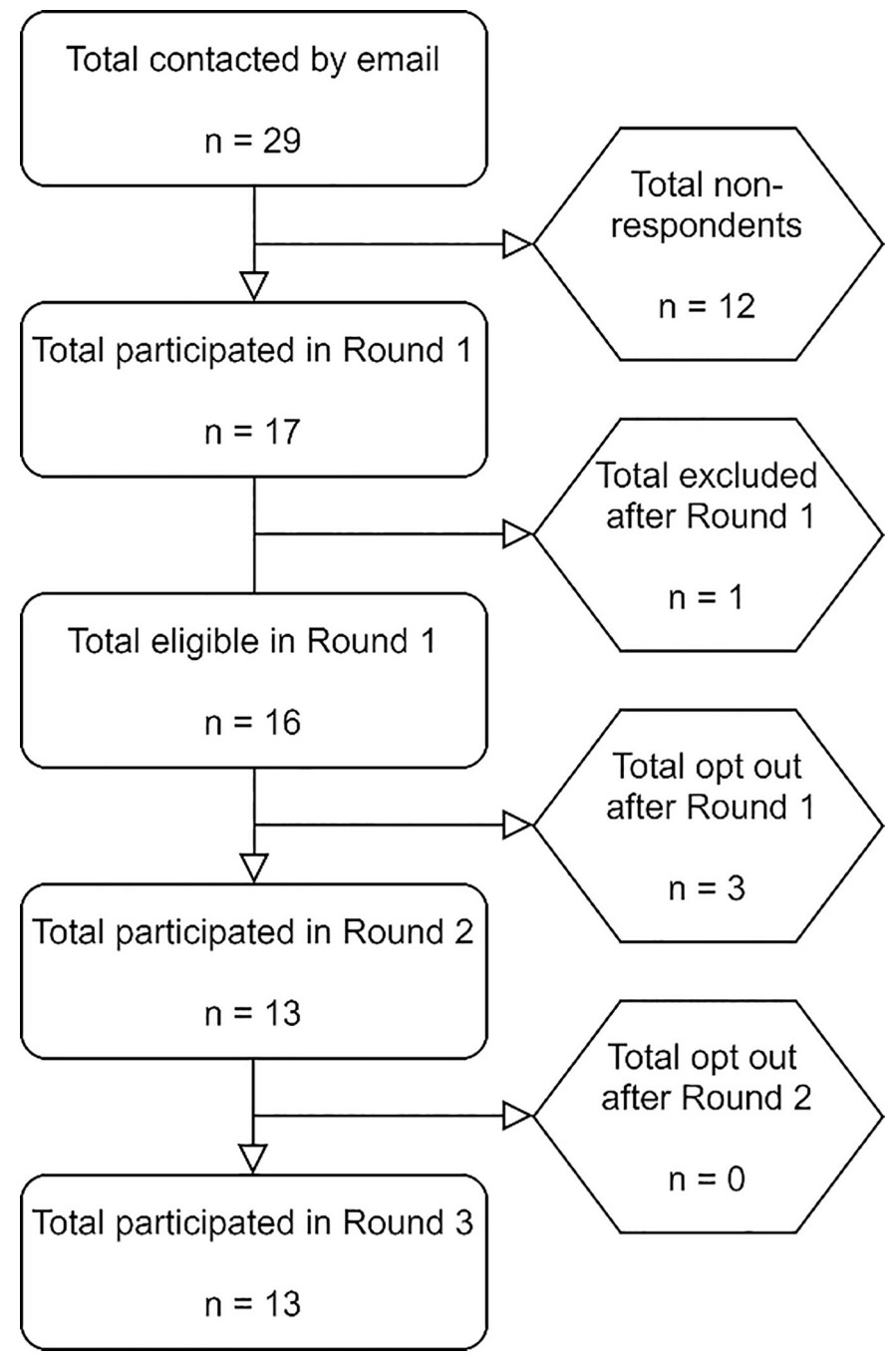

**Fig 1. Flowchart.**

## Themes

**Primary care workforce.** Informaticians disagreed with the prediction that in the US "there is a 90% chance primary care doctors will be obsolete 100 years from now". Panelists also agreed that primary care in the US would be one of the last specialties to be replaced by AI/ML. In the short-term, by 2029 in the US, experts forecast that advancements in AI/ML will incur a number of workforce changes in primary care (see Fig 2).

**Table 3. IQR, mean, and SD for Round 2 & 3.**

| Projection | Scale [1] | Round 2 (n = 13) | | | Round 3 (n = 13) | | | Change | |
|---|---|---|---|---|---|---|---|---|---|
| | | IQR | m | (SD) | IQR | m | (SD) | m | (SD) |
| **Access to Care** | | | | | | | | | |
| 28. the use of telemedicine | I | 1 | **6.23** | (.60) | - | - | - | - | - |
| 32. AI/ML will be used for appointment scheduling | L | 2 | 5.77 | (1.42) | 1 | **6.23** | (.73) | .46 | (-.69) |
| 27. the use of AI/ML patient triage tools by health organization | I | 1 | **5.62** | (.65) | - | - | - | - | - |
| 26. as a result of AI/ML tools, basic medical expertise via electronic devices | I | 1 | **5.54** | (.88) | - | - | - | - | - |
| 24. as a result of AI/ML tools, patient access to medical care[2] | I | 0 | **5.08** | (.86) | - | - | - | - | - |
| 25. as a result of AI/ML tools, patient access to expert doctor knowledge[2] | I | 0 | **4.92** | (.64) | - | - | - | - | - |
| 31. AI/ML will be used for patient-doctor matching | L | 2 | 4.62 | (1.56) | 1 | **4.85** | (1.34) | .23 | (-.22) |
| 29. when it comes to the impact of AI/ML on patient access to medical care the US will lag behind other developed countries | L | 2 | 3.85 | (1.63) | 2 | 3.77 | (1.59) | -.08 | (-.04) |
| 30. AI/ML enabled resources will be too expensive for most patients | L | 2 | 3.00 | (1.35) | 2 | 3.46 | (1.61) | .46 | (.26) |
| **Diagnostic Accuracy** | | | | | | | | | |
| 3. diagnostic accuracy for some conditions where the markers of illness are relatively homogenous | I | 1 | **5.69** | (.85) | - | - | - | - | - |
| 8. AI/ML-enabled tools will be routinely used to assist primary care doctors with diagnosing the most difficult cases | L | 2 | 5.54 | (1.39) | 1 | **5.62** | (1.04) | .08 | (-.35) |
| 1. diagnostic accuracy | I | 1 | **5.54** | (.88) | - | - | - | - | - |
| 7. AI/ML-enabled tools will be routinely used to assist doctors in diagnostic decision-making | L | 1 | **5.46** | (1.39) | - | - | - | - | - |
| 4. diagnostic accuracy for rare conditions | I | 1 | **5.38** | (.87) | - | - | - | - | - |
| 9. AI/ML-enabled tools will be routinely used by patients to self-diagnose | L | 1 | **5.38** | (1.26) | - | - | - | - | - |
| 10. revamped nosology of many symptom-based disease categories | L | 1 | **5.31** | (.95) | - | - | - | - | - |
| 2. diagnostic accuracy for minority patients | I | 2 | 5.15 | (.90) | 1 | **5.08** | (.95) | -.07 | (.05) |
| 5. rates of over-diagnosis | I | 2 | 3.85 | (1.34) | 2 | 4.00 | (1.22) | .15 | (-.12) |
| 6. rates of unnecessary testing | I | 2 | 3.77 | (1.36) | 2 | 3.69 | (1.32) | -.08 | (.40) |
| **Healthcare Disparities** | | | | | | | | | |
| 14. private hospitals will have an advantage in using AI/ML resources to improve diagnostic accuracy compared to public hospitals | L | 2 | 4.77 | (1.09) | 0 | **5.00** | (.82) | .23 | (-.27) |
| 13. AI/ML tools will improve diagnostic accuracy for those with limited access to human experts | L | 2 | 4.92 | (1.04) | 1 | **5.00** | (.91) | .08 | (-.13) |
| 15. there will be representative data collection among minority groups | L | 2 | 4.54 | (1.33) | 0 | **5.00** | (1.00) | .46 | (-.33) |
| 12. more sophisticated AI/ML resources will only be available to higher income individuals | L | 2 | 4.77 | (1.83) | 2 | 4.54 | (1.56) | -.23 | (-.27) |
| 11. as a result of AI/ML enabled tools, healthcare disparities | I | 0 | **4.15** | (.80) | - | - | - | - | - |
| **Empathic Care of Patients** | | | | | | | | | |
| 21. AI/ML tools will help assist doctors in shared decision-making with patients | L | 1 | **5.54** | (1.05) | - | - | - | - | - |
| 22. AI/ML tools will help clinicians to think more about patients' lifestyle | L | 2 | 5.31 | (1.03) | 1 | **5.38** | (1.12) | 0 | (.09) |
| 20. health care will be increasingly productized | L | 2 | 4.85 | (1.68) | 1 | **5.23** | (.93) | .38 | (-.75) |
| 23. AI/ML tools will use data on the social determinants of health to devise personalized health plans | L | 1 | **5.15** | (1.21) | - | - | - | - | - |
| 16. the availability of AI/ML tools mean that levels of empathic care | I | 1 | **4.62** | (.87) | - | - | - | - | - |
| 17. the availability of AI/ML tools mean the total time patients spend with doctors | I | 0 | **4.00** | (.71) | - | - | - | - | - |
| 18. the availability of AI/ML tools mean the documentation burden on doctors | I | 1 | **3.62** | (.77) | - | - | - | - | - |
| 19. AI/ML will offer direct resources for delivering empathic care | L | 3 | 3.46 | (1.71) | 2 | 3.31 | (1.49) | -.15 | (-.22) |
| **Primary Care Workforce & Its Long-term Future** | | | | | | | | | |
| 49. adoption of AI/ML tools in health care will be slow due to the culture of medicine | A | 3 | 5.23 | (1.88) | 2 | 5.69 | (1.32) | .46 | (-.56) |
| 34. the number of clinicians with degrees in engineering or computer science entering medicine | I | 1 | **5.54** | (.88) | - | - | - | - | - |
| 33. the proportion of mid-level clinicians (e.g. nurse practitioners) | I | 1 | **5.54** | (1.05) | - | - | - | - | - |
| 39. AI/ML tools will change the reimbursement structure for routine clinical tasks | L | 2 | 5.00 | (1.73) | 1 | **5.54** | (1.13) | .54 | (-.6) |
| 44. primary care doctors will be one of the last specialties to be replaced by AI/ML in medicine | A | 1 | **5.46** | (1.51) | - | - | - | - | - |
| 47. primary care doctors will always be required to deliver empathic aspects of care | A | 2 | 5.46 | (1.94) | 2 | 5.46 | (1.90) | 0 | (-.04) |
| 36. training requirements in working with AI/ML | I | 1 | **5.23** | (1.01) | - | - | - | - | - |

*(Continued)*

**Table 3.** (Continued)

| Projection | Scale [1] | Round 2 (*n* = 13) | | | Round 3 (*n* = 13) | | | Change | |
|---|---|---|---|---|---|---|---|---|---|
| | | IQR | m | (SD) | IQR | m | (SD) | m | (SD) |
| 35. efficiency in the delivery of primary care | I | 1 | **5.08** | (.76) | - | - | - | - | - |
| 46. primary care doctors will always be required to synthesize information [4] | A | 1 | **5.08** | (1.38) | - | - | - | - | - |
| 45. primary care doctors will always be required as gatekeepers in medicine [4] | A | 3 | 4.62 | (1.66) | 1 | **4.85** | (1.57) | .23 | (-.09) |
| 38. doctors will transition from the role of dispensers of knowledge to managing teams and information systems | L | 2 | 4.31 | (1.93) | 3 | 4.69 | (1.60) | 0 | (-1) |
| 37. AI/ML tools will enable clinicians with lower licenses to do higher-level jobs | L | 3 | 4.31 | (1.65) | 1 | **4.54** | (1.39) | .23 | (-.26) |
| 48. patients will always prefer humans as gatekeepers of their medical care | A | 3 | 4.46 | (1.85) | 3 | 4.38 | (1.80) | -.08 | (-.05) |
| 51. there is a 90% chance that primary care doctors will be obsolete 100 years from now | A | 2 | 2.62 | (1.98) | 1 | **2.77** | (1.88) | .15 | (-.1) |
| 50. there is a 90% chance that primary care doctors will be obsolete 50 years from now | A | 2 | 2.46 | (2.03) | 2 | 2.46 | (2.03) | 0 | (0) |
| *Technological Advancements in Primary Care* | | | | | | | | | |
| 43. regulatory issues in improving diagnostic accuracy of AI/ML tools will be more challenging than technical issues | A | 2 | 5.85 | (.99) | 1 | **5.54** | (1.39) | -.31 | (.4) |
| 40. Will improvements in the diagnostic accuracy of AI/ML tools require technological breakthroughs? | Y/N | - | 7 [3] | (54%) | - | 3 [3] | (23%) [3] | -4 | -31% |
| 40a. If 'Yes', will require technological breakthroughs in causal modelling | A | 1 | 6.43 | (.79) | - | - | - | - | - |
| 40b. If 'Yes', will require technological breakthroughs in artificial general intelligence | A | 2 | 5.29 | (1.50) | 3 | 4.78 | (1.64) | -.51 | (.14) |
| 40c. If 'Yes', will require technological breakthroughs in the interpretability of certain approaches such as deep learning | A | 1 | 5.57 | (1.72) | - | - | - | - | - |
| 40d. If 'Yes', will require technological breakthroughs in human-level natural language processing | A | 0 | 6.71 | (.76) | - | - | - | - | - |
| 42. will require integrated data sets | A | 1 | 6.69 | (.48) | - | - | - | - | - |
| 40e. If 'Yes', will require technological breakthroughs in semi-supervised learning | A | 1 | 6.43 | (.79) | - | - | - | - | - |
| 40f. If 'Yes', will require technological breakthroughs to harness the sensor data from smartphones and wearables to forecast individual symptom trajectories. | A | 1 | 6.00 | (1.83) | - | - | - | - | - |
| 41. will require improved data quality | A | 1 | 6.15 | (1.63) | - | - | - | - | - |

[1] Scales were either *increase* (I)– 1. greatly decrease to 7. greatly increase, *likelihood* (L)– 1. very unlikely to 7. very likely, or *agreement* (A)– 1. strongly disagree to 7. strongly agree.

[2] While there is some ambiguity between how to interpret the difference between these statements, we retained them to preserve the predictions as submitted by our experts.

[3] Count and percentage of 'Yes' responses.

[4] These items were later omitted since, on further reflection it as unclear what might be meant by the term "required".

Values in bold indicate consensus statements. *IQR*–interquartile range, *m*–mean, *SD*–standard deviation.

Dependent on educational background some contrasting predictions emerged (see Table 4). Beyond 2029, experts without a medical degree (MD) considered it likely that primary care doctors would always be needed to deliver empathic aspects of care–a prediction

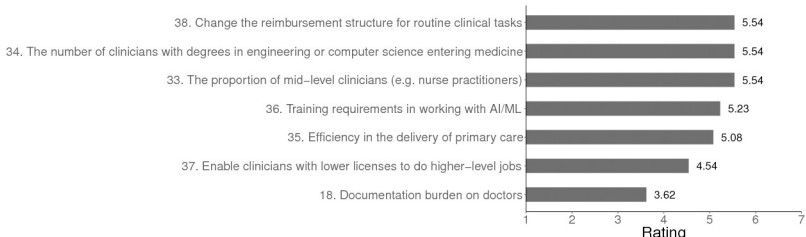

**Fig 2. Predicted changes in the US primary care workforce in 2029.** *Note.* For items 33, 34, 35, 36 the scale used was: *Greatly decrease* = 1, *Remain the same* = 4, *Greatly increase* = 7. For items 37 and 39: *Very unlikely* = 1, *Uncertain* = 4, *Very likely* = 7.

**Table 4. Table of statements that did not reach consensus & statistics based on educational background.**

| *Projection* | Scale [1] | Medical (*n* = 6) [2] | | | Non-Medical (*n* = 6) [†] | | |
|---|---|---|---|---|---|---|---|
| | | *IQR* | *m* | *(SD)* | *IQR* | *m* | *(SD)* |
| *Access to Care* | | | | | | | |
| 29. when it comes to the impact of AI/ML on patient access to medical care the US will lag behind other developed countries | L | 1.75 | 3.17 | (2.14) | **1** | **4.50** | (.55) |
| 30. AI/ML enabled resources will be too expensive for most patients | L | 2 | 3.33 | (2.16) | 1.75 | 3.67 | (1.21) |
| *Diagnostic Accuracy* | | | | | | | |
| 5. rates of over-diagnosis | I | 2.25 | 3.67 | (1.37) | 1.5 | 4.00 | (.89) |
| 6. rates of unnecessary testing | I | 1.5 | 3.17 | (1.17) | 1.5 | 3.83 | (1.17) |
| *Healthcare Disparities* | | | | | | | |
| 12. more sophisticated AI/ML resources will only be available to higher income individuals | L | 2.5 | 3.67 | (1.75) | 1.75 | 5.17 | (.98) |
| *Empathic Care of Patients* | | | | | | | |
| 19. AI/ML will offer direct resources for delivering empathic care | L | **0.75** | **3.33** | (1.21) | 1.75 | 3.00 | (1.79) |
| *Primary Care Workforce & Its Long-term Future* | | | | | | | |
| 38. doctors will transition from the role of dispensers of knowledge to managing teams and information systems | L | **0.75** | **5.00** | (1.10) | 1.75 | 4.00 | (1.79) |
| 47. primary care doctors will always be required to deliver empathic aspects of care | A | 1.5 | 5.50 | (1.87) | **0.75** | **5.17** | (2.14) |
| 48. patients will always prefer humans as gatekeepers of their medical care | A | 2.5 | 4.50 | (1.87) | 2.25 | 4.00 | (1.90) |
| 49. adoption of AI/ML tools in health care will be slow due to the culture of medicine | A | 2.5 | 5.33 | (1.63) | **0.75** | **6.00** | (1.10) |
| 50. there is a 90% chance that primary care doctors will be obsolete 50 years from now | A | 2.5 | 2.50 | (2.07) | 1.75 | 2.50 | (2.35) |
| *Technological Advancements in Primary Care* | | | | | | | |
| 40b. If 'Yes', will require technological breakthroughs in artificial general intelligence | A | 1.75 | 5.25 | (1.71) | 1.75 | 4.25 | (1.89) |

[1] Scales were either *increase* (I)– 1. greatly decrease to 7. greatly increase, *likelihood* (L)– 1. very unlikely to 7. very likely, and *agreement* (A)– 1. strongly disagree to 7. strongly agree.

[2] One respondent had both medical and technological backgrounds so their data was excluded from both samples.

*IQR*–interquartile range, *m*–mean, *SD*–standard deviation.

that did not engender consensus among panelists with a medical degree. Similarly, panelists without a medical education strongly agreed that the adoption of AI/ML tools in US healthcare will be slow, by 2029, due to the culture of medicine while those with a medical education did not reach consensus on this item. Conversely, experts with a medical degree forecast that by 2029 US doctors will transition from the role of dispensers of knowledge to managing teams and information systems; however, there was no consensus on this item among participants without an MD.

**Diagnostic accuracy.** Overall, our experts forecast that by 2029 AI/ML will increase rates of diagnostic accuracy especially for conditions where the markers of illness are relatively homogenous (see Fig 3). Among their predictions, panelists envisaged that by AI/ML tools will improve diagnostic accuracy among persons with limited access to human experts, individuals identifying as from minority groups, or for those with rare conditions.

**Access to care.** As a result of the disintermediation of physicians' expertise, our experts predicted that by 2029, AI/ML will increase access to primary care in the US (see Fig 4).

Comparisons of ratings between participants with and without a medical education resulted in some divergence. Respondents without an MD predicted that, by 2029 as a result of AI/ML, patient access to medical care in the US will lag behind other developed countries; participants with an MD did not reach consensus on this item (see Table 4).

**Empathic care of patients.** Experts envisaged that by 2029 in the US, the availability of AI/ML tools will help to augment levels of empathic care (see Fig 5).

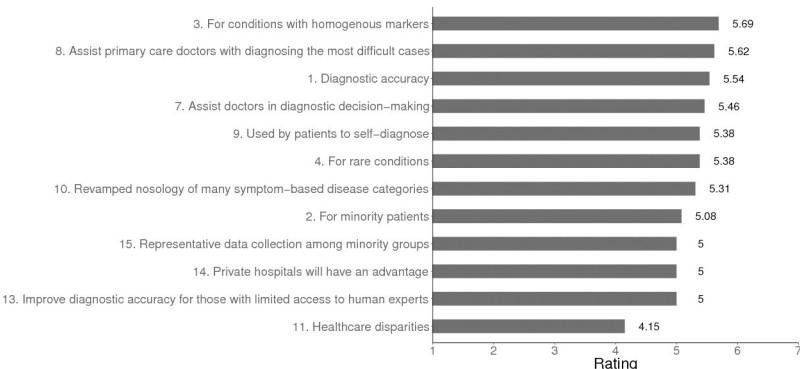

**Fig 3. Predicted changes in diagnostic accuracy in US primary care in 2029.** *Note*. For items 1, 2, 3, 4, 11 the scale used was: *Greatly decrease* = 1, *Remain the same* = 4, *Greatly increase* = 7. For items 7, 8, 9, 10, 13, 14, 15 the scaled used was: *Very unlikely* = 1, *Uncertain* = 4, *Very likely* = 7.

Panelists were divided on whether, by 2029, AI will offer direct resources for delivering empathic care to patients (see Table 4). Participants with an MD considered this unlikely, while others failed to reach consensus on this item.

## Discussion

### Summary of major findings

The collective forecasts of medical informaticians have been missing from discussions about how AL/ML will influence the short-term future of primary care (see Box 1). In this Delphi poll there was consensus that in the next decade in the US, AI/ML will engender training and primary care work forces changes, improve rates of diagnostic accuracy, and increase access to primary care.

Economists forecast that in the coming decades, AI/ML will revolutionize the workplace [35, 36]. Taking a long view, informaticians in this Delphi poll predicted that 100 years from now it is unlikely that primary care doctors will be obsolete. Panelists further envisaged that primary care will be one of the last medical specialties to be displaced by technology. However, in the short term, by 2029, our experts did foresee workforce and training changes in US primary care as a result of AI/ML. Experts were collectively uncertain about whether AI/ML tools would enable lower-level clinicians to do higher level jobs, though it was not clear whether this prediction was driven by technological or regulatory considerations. Panelists anticipated a shift towards computing and engineering in the educational background of students entering medical school in 2029, and increased training demands on medical students to work with AI/

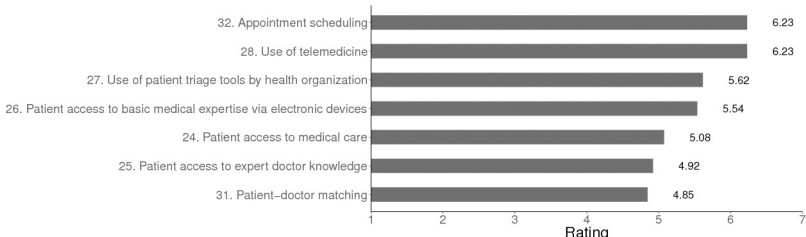

**Fig 4. Predicted changes in access to US primary care in 2029.** *Note*. For items 24, 25, 26, 27, 28 the scale used was: *Greatly decrease* = 1, *Remain the same* = 4, *Greatly increase* = 7. For items 31 and 32 the scale used was: *Very unlikely* = 1, *Uncertain* = 4, *Very likely* = 7.

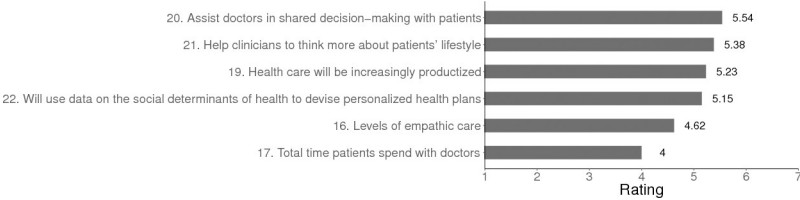

**Fig 5. Predicted changes in empathic care of patients in US primary care in 2029.** *Note.* For items 16, 17, 18 the scale used was: *Greatly decrease* = 1, *Remain the same* = 4, *Greatly increase* = 7. For items 20, 21, 22, 23: *Very unlikely* = 1, *Uncertain* = 4, *Very likely* = 7.

ML in healthcare. However, the survey did not reveal whether experts perceived there to be risks of physicians using ML/AL tools without a computing and engineering background, or indeed, an ethics or evidence-based perspective, on these techniques. The panel's predictions on education trends should also be observed against the currently limited debate about the need for curricular changes in medical education [37–39].

For many reasons, including financial, social, and geographical, timely access to primary care in the US remains a considerable problem. Compounding matters, with fewer medical students entering primary care, inefficiencies, and demographic changes–an ageing population, and more people suffering from chronic conditions for longer–it is widely envisaged that ambulatory medicine will become increasingly strained [40, 41]. The results of this Delphi poll suggest that AI/ML tools may help to address some of these challenges. Experts envisaged that by 2029 there would be increased access to care via AI/ML-enabled tools for medical triage and routine patient self-diagnosis, and with the growth of telemedicine.

The panel also predicted increasing medical precision. By 2029, experts envisaged that the use of AI/ML-enabled tools among patients will help to reduce diagnostic errors both for diseases with homogenous symptoms, and for more difficult medical cases. Perhaps contributing to these reductions, experts anticipated that advancements in AI/ML will engender revisions in disease classifications. These positive predictions should be viewed against current evidence that diagnostic error is both common and harmful. In the US, recent estimates suggest a diagnostic error rate of 13–15% affecting the lives of around 12 million Americans annually, contributing to 10% of all deaths, and the highest proportion of medical malpractice claims [42–44]. Patients from racial and ethnic minorities, and those on low-incomes, are at higher risk of diagnostic error [45]. Our experts predicted that diagnostic accuracy will increase for individuals with limited access to care, minorities, or patients with rare conditions. While there are currently considerable concerns about the potential for algorithmic biases to be baked into AI/ML tools, driven in part by the underrepresentation of underprivileged demographic groups in training phases of machine learning [46], there was consensus among our Delphi panel that data collection in 2029 will be more representative of minority populations. This prediction may help to explain why the panel anticipated improved diagnostic accuracy for minorities.

Nonetheless, experts were less optimistic that AI/ML will narrow health disparities in the US by 2029. Current findings point to a "digital divide" in healthcare. Many factors drive current differential usage of digital health innovations including costs, lack of broadband access, and lower levels of digital and health literacy among underprivileged populations [47, 48]. Research also suggests that in the US, health app usage is more common among people who are younger, better educated, on a higher income, or in better health [49]. Our panel predicted that US healthcare will become increasingly productized. Although the poll provided no causal explanations for this prediction, in a growing health app economy, experts may have anticipated that disadvantaged patients will continue to be less likely to adopt eHealth tools. In

addition, there was consensus that private hospitals will have greater access to AI/ML-enabled resources to improve diagnostic accuracy than public hospitals. Existing structural disparities in care may also have been perceived to be a factor that will perpetuate inequities in eHealth.

Our Delphi poll provided nuanced forecasts on the theme of physician empathy. There was collective consensus that, by 2029, AI/ML would not free up more time with patients in US primary care; however, experts did forecast that levels of empathy in primary care would increase in this time period. The survey did not fully illuminate the reasons for this but there was consensus that AI/ML-enabled tools will assist physicians in shared decision-making, and help provide information on patients' lifestyles and the social determinants of individuals' health. Conceivably, the panel may have envisaged that such data might enhance physicians' personal knowledge about patients thereby fostering more empathic care. Again, these views appear to differ subtly from those of physicians. In qualitative research a common prediction among physicians is that, by liberating health professionals from administrative tasks, AI/ML will indirectly facilitate more time with patients thereby enhancing levels of empathy [11, 15] Survey research also indicates skepticism among physicians that AI/ML will be able to directly substitute for, or augment clinicians, in the provision of empathic care [10–12, 15].

In terms of physicians' responsibilities, experts did not envisage that AI/ML will help to reduce documentation burdens by 2029 [10, 12]. This prediction contrasts with the more optimistic opinions of surveyed physicians. For example, in 2019, a global survey of psychiatrists found that the majority (83%, 657/791) judged it likely that future technology will fully replace physicians in the task of documentation with 84% (552/657) of these respondents predicting that this will happen in the next 10 years [12]. Similarly, in 2018, survey research conducted among primary care physicians in the UK revealed comparable results: most UK general practitioners (80%, 578/720) anticipated that future technology will fully replace humans in the task of documentation with 79% (458/578) of these respondents believing that this will happen in the next decade [10].

Finally, experts in this Delphi poll did not weigh in on specific policy, legal, or ethical issues in relation to the impact of AI/ML on primary care. However, there was consensus that by 2029 regulatory issues will pose greater challenges than technical problems.

## Strengths and limitations

To our knowledge this is the first Delphi poll to explore experts' predictions about the short-term effects of AI/ML on a medical specialism. Major strengths of the survey were the high response rates between rounds, and the diversity of participants. Although only around one third of Round 3 panelists were female (31%), currently around 25% of health IT leaders in the US are women [50]. The expert panel comprised leading health informaticians around half of whom also had a medical background. Panelists were drawn from diverse backgrounds, nationalities, and ethnicities including 3 participants in Round 1 who do not reside in the US but who are knowledgeable about the US healthcare system. We also note that the majority of experts primarily held allegiances to academia, and medicine, rather than industry; nonetheless, this may have been a strength rather than a limitation, resulting in more modest predictions.

This survey has several limitations. As with all Delphi polls, there is no guarantee of accuracy in forecasts. No standardized guidelines exist for identifying, excluding, or selecting suitable experts from the field of interest [23, 27]. Reliability of predictions is dependent on the specialist knowledge of the participants which can be influenced by norms and values, motivational biases, and stakeholder interests [51, 52]. Although there was strong consensus among our panel of experts, we noted some divergence in opinions between participants with and

without a medical degree. Conceivably, professional medical allegiances may have affected predictions; overall, however, we cannot speculate on how the composition of our panel strengthened or diminished the quality of predictions. Whilst participant retention rates between rounds were high, the number of panelists was limited, and more participants in the first round may have resulted in different consensus opinions [53, 54].

Importantly, two events arising in the immediate period after data collection—one global and one in the US context—may affect the reliability of the Delphi poll. The coronavirus pandemic has (and currently is) exerting a significant impact on the delivery of primary care in the US. Driven by this crisis, current evidence shows a substantial uptick in demand for telemedicine consultations, and in the use of AI/ML-driven triage tools [55, 56]. Although it is too early to predict with certainty whether increase in these applications will persist after the pressure on frontline medicine has abated, it seems possible that our experts' forecasts on the influence of AI/ML on access to care may be especially well supported.

Second, the survey was administered prior to the finalized ruling, in March 2020, by the National Coordinator of Health Information Technology (ONC) on the 21st Century Cures Act [57]. Designed to maximize innovation in healthcare by creating a competitive health app economy, this federal ruling sets out technical standards about how data must be shared, mandating patients' right to access their digital medical records. While the final ruling may have been anticipated by some of our experts in the months preceding the announcement, we cannot be certain about whether or how its publication might otherwise have influenced consensus predictions of our participants. However, we suggest that uncertainty prior to the ruling may have fostered more cautious predictions about the impact of AI/ML on primary care among our experts.

## Conclusions

*A good hockey player plays where the puck is. A great hockey player plays where the puck is going to be.*

- Wayne Gretzky

This Delphi poll provides the consensus predictions of leading health informaticians on the impact of AI/ML on primary care in the US. The panel forecast that, in the long-term (100 years from now) primary care doctors will not be obsolete, and furthermore, that general medicine will be one of the last medical specialties to be displaced by technology. By 2029 in the US, however, experts did forecast that AI/ML will exert an impact on the delivery and quality of primary care. Specifically, the panel predicted increased rates of diagnostic accuracy including for the most disadvantaged patient populations, greater access to primary care, and enhanced levels of empathic patient care. Against the panel's forecast that healthcare in the US would be increasingly productized, there was consensus that regulatory issues will pose greater challenges than technical ones in improving diagnostic accuracy. Experts were also less optimistic about the prospects of AI/ML to precipitate other desirable short-term changes in medicine. By 2029 in the US, the panel predicted that AI/ML would *not* narrow healthcare disparities, reduce documentation burdens on primary care physicians, or increase the total time spent with patients. In the next decade, experts forecast increased AI/ML training requirements for medical students.

The central goal of Delphi polls is expert prediction. However, forecasts can also help us to exert control over the future by facilitating forward planning, and focusing attention on where, and how, relevant actors might intervene to create preferable outcomes. Innovations in

digital care pose myriad practical, ethical, and regulatory issues including (but by no means limited to): the creations of standards for assessing the reliability and approval of medical algorithms and apps, questions about patient privacy, and the security of patients' online health information [58, 59]. In reviewing these findings we are struck by the contrastive predictions of our experts with those of surveyed physicians [10, 11]. As others have noted, medical schools have been slow to adapt curricula and offer courses aimed at promoting AI/ML literacy among students [37–39]. We conclude that to empower both physicians and patients, and to rise to the challenges of the next decade, it is incumbent on the medical community, health and medical educators, and policy-makers to take action to improve digital literacy both among patients and our current and future health professionals [59].

## Supporting information

**S1 Appendix. Survey Round One.**
(PDF)

**S2 Appendix. Survey Round Two.**
(PDF)

**S3 Appendix. De-identified raw data results of Round Two.**
(PDF)

## Acknowledgments

It is standard practice in Delphi polls to acknowledge the panel of experts who shared their valuable insights. We are indebted to them for giving up their time. The following participants (12 out of 16) granted us permission to report their names: David Bates, MD; Andrew Beam, PhD; Gabriel Brat, MD; Wendy Chapman, PhD; Enrico Coiera, PhD FACMI; Peter Embi, MD, MS, FACP, FACMI, FAMIA; John Halamka, MD; Arjun Manrai, PhD; JP Onella, PhD; Neil Sebire, MBBS, BClinSCi, MD, FRCOG, FRCPath, FFCI; Jessie Tenenbaum, PhD; Kun-Hsing Yu, MD PhD.

## Author Contributions

**Conceptualization:** Charlotte Blease.

**Data curation:** Anna Kharko.

**Formal analysis:** Charlotte Blease, Anna Kharko, Cosima Locher.

**Investigation:** Charlotte Blease, Anna Kharko, Kenneth D. Mandl.

**Methodology:** Charlotte Blease, Anna Kharko, Cosima Locher, Catherine M. DesRoches, Kenneth D. Mandl.

**Resources:** Anna Kharko.

**Software:** Anna Kharko.

**Supervision:** Catherine M. DesRoches, Kenneth D. Mandl.

**Visualization:** Anna Kharko.

**Writing – original draft:** Charlotte Blease.

**Writing – review & editing:** Charlotte Blease, Anna Kharko, Catherine M. DesRoches, Kenneth D. Mandl.

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
