## [Decision Letter · Decision Letter 0]

28 Aug 2020

PONE-D-20-12979

US Primary Care in 2029: A Delphi Survey on the Impact of Machine Learning

PLOS ONE

Dear Dr.ssa Blease,

Thank you for submitting your manuscript to PLOS ONE. After careful consideration, we feel that it has merit but does not fully meet PLOS ONE’s publication criteria as it currently stands. Therefore, we invite you to submit a revised version of the manuscript that addresses the points raised during the review process.

We look forward to receiving your revised manuscript.

Kind regards,

Claudia Marotta

Academic Editor

PLOS ONE

Journal Requirements:

2.We note that you have indicated that data from this study are available upon request. PLOS only allows data to be available upon request if there are legal or ethical restrictions on sharing data publicly. For information on unacceptable data access restrictions, please see http://journals.plos.org/plosone/s/data-availability#loc-unacceptable-data-access-restrictions.

4. Please include a caption for figure 3. (Figure 2 caption included 2x)

Additional Editor Comments (if provided):

Dear Authors,

I appreciate a lot your manuscript

Follow reviewer indications you can improve your article

Reviewers' comments:

Reviewer's Responses to Questions

**Comments to the Author**

1. Is the manuscript technically sound, and do the data support the conclusions?

Reviewer #1: Yes

Reviewer #2: Yes

2. Has the statistical analysis been performed appropriately and rigorously? 

Reviewer #1: Yes

Reviewer #2: Yes

3. Have the authors made all data underlying the findings in their manuscript fully available?

Reviewer #1: Yes

Reviewer #2: Yes

4. Is the manuscript presented in an intelligible fashion and written in standard English?

Reviewer #1: Yes

Reviewer #2: Yes

5. Review Comments to the Author

Reviewer #1: Congratulations for the appropriate methodology and for the clarity of your explanation.

Probably the only integration that can be useful in the discussion is the specification of why the shift towards computing and engineering in the educational background of students entering medical school is considered necessary:

- are there any risks if these techniques, which will become standards of care soon, are used in the absence of specific training? If there is any relevant literature please specify, if not it will be an interesting perspective to be pursued by further study.

Reviewer #2: Blease et al used a modified Delphi poll to solicit health informaticians’ predictions about the impact of AI/ML on primary care in US in 2029. The text is well-written and the flow is logical. The experiments are well-planned and executed. While there are many interesting questions that arise from this work, e.g. would there be contrasting predictions between medical experts with and without computer science background, the manuscript in its current form tells a coherent story and build a foundation for future work. One minor comment - I think the authors should cite Liyanago et al. (https://www.ncbi.nlm.nih.gov/pmc/articles/PMC6697547/)

6. PLOS authors have the option to publish the peer review history of their article (what does this mean?). If published, this will include your full peer review and any attached files.

Reviewer #1: **Yes: **Calogero Casà

Reviewer #2: No

---

## [Author Response · Author response to Decision Letter 0]

3 Sep 2020

31 August 2020

Dear Plos One,

Thank you for the excellent feedback and for considering a resubmission of our paper. Please find our response to the reviewers and editor of our manuscript, below.

PONE-D-20-12979

US Primary Care in 2029: A Delphi Survey on the Impact of Machine Learning

PLOS ONE

Response to Editor 

Response: Round 1 of the Delphi Poll includes demographic data, and qualitative responses. Because of the sample size, even with de-identification of this data, it would be possible to re-identify the participants thereby posing a risk to respondents’ anonymity. In addition, Round 3 of the Delphi Poll comprised 13 bespoke, individual surveys to each participant. Again, this information cannot be shared publicly because of the need to ensure participant anonymity and confidentiality, as per ethical review. However, de-identified raw data is now available as a supplementary files Appendix 3 for Round 2 of the survey.

Response: Please see above.

Response: now added.

4. Please include a caption for figure 3. (Figure 2 caption included 2x)

 Response: We were unsure about what was being requested. Caption for Fig 3 is now included and highlighted in the text. Please let us know if any further amendments are required. Fig 3. Predicted changes in diagnostic accuracy in US primary care in 2029

Additional Editor Comments (if provided):

Dear Authors,

I appreciate a lot your manuscript

Follow reviewer indications you can improve your article

Response: Thank you.

Reviewers' comments:

Reviewer's Responses to Questions

Comments to the Author

1. Is the manuscript technically sound, and do the data support the conclusions?

Reviewer #1: Yes

Reviewer #2: Yes

2. Has the statistical analysis been performed appropriately and rigorously?

Reviewer #1: Yes

Reviewer #2: Yes

3. Have the authors made all data underlying the findings in their manuscript fully available?

Reviewer #1: Yes

Reviewer #2: Yes

4. Is the manuscript presented in an intelligible fashion and written in standard English?

Reviewer #1: Yes

Reviewer #2: Yes

5. Review Comments to the Author

Reviewer #1: Congratulations for the appropriate methodology and for the clarity of your explanation.

Probably the only integration that can be useful in the discussion is the specification of why the shift towards computing and engineering in the educational background of students entering medical school is considered necessary:

- are there any risks if these techniques, which will become standards of care soon, are used in the absence of specific training? If there is any relevant literature please specify, if not it will be an interesting perspective to be pursued by further study.

Response: We thank Reviewer 1 for these kind comments. Great point. While there is no relevant literature on the matter, we have now added the following nuanced point on page 22:

“However, the survey did not reveal whether experts perceived there to be risks of physicians using ML/AL tools without a computing and engineering background, or indeed, without an ethics or evidence-based perspective, on these techniques. The panel’s predictions on education trends should also be observed against the currently limited debate about the need for curricular changes in medical education.(37–39)”.

In addition, to draw out the ethical, as opposed to technical considerations, in the Conclusion we have added 2 additional citations on ethics and AI, to reflect the importance of ethical education to digital literacy among clinicians and the public:

Vayena E, Blasimme A, Cohen IG. Machine learning in medicine: Addressing ethical challenges. PLoS medicine. 2018;15(11):e1002689.

Zuboff S. The Age of Surveillance Capitalism: The Fight for a Human Future at the New Frontier of Power. Profile Books; 2019.

Reviewer #2: Blease et al used a modified Delphi poll to solicit health informaticians’ predictions about the impact of AI/ML on primary care in US in 2029. The text is well-written and the flow is logical. The experiments are well-planned and executed. While there are many interesting questions that arise from this work, e.g. would there be contrasting predictions between medical experts with and without computer science background, the manuscript in its current form tells a coherent story and build a foundation for future work. One minor comment - I think the authors should cite Liyanago et al. (https://www.ncbi.nlm.nih.gov/pmc/articles/PMC6697547/)

Response: We thank Reviewer 2 for this encouraging feedback. We have now included the citation in the paper [citation 17]. On page 4 we have amended the text as follows:

“Currently, there is scarce exploration of consensus views among informaticians (17); in particular, on how AI/ML might meaningfully influence medical care in the short-term (18).”

6. PLOS authors have the option to publish the peer review history of their article (what does this mean?). If published, this will include your full peer review and any attached files.

Do you want your identity to be public for this peer review? For information about this choice, including consent withdrawal, please see our Privacy Policy.

Reviewer #1: Yes: Calogero Casà

Reviewer #2: No

---

## [Decision Letter · Decision Letter 1]

16 Sep 2020

US Primary Care in 2029: A Delphi Survey on the Impact of Machine Learning

PONE-D-20-12979R1

Dear Dr. Blease,

We’re pleased to inform you that your manuscript has been judged scientifically suitable for publication and will be formally accepted for publication once it meets all outstanding technical requirements.

Kind regards,

Claudia Marotta

Academic Editor

PLOS ONE

Additional Editor Comments (optional):

Dear Authors, congratulations for your great article!

Reviewers' comments:

Reviewer's Responses to Questions

**Comments to the Author**

1. If the authors have adequately addressed your comments raised in a previous round of review and you feel that this manuscript is now acceptable for publication, you may indicate that here to bypass the “Comments to the Author” section, enter your conflict of interest statement in the “Confidential to Editor” section, and submit your "Accept" recommendation.

Reviewer #1: All comments have been addressed

Reviewer #2: All comments have been addressed

2. Is the manuscript technically sound, and do the data support the conclusions?

Reviewer #1: Yes

Reviewer #2: Yes

3. Has the statistical analysis been performed appropriately and rigorously? 

Reviewer #1: Yes

Reviewer #2: Yes

4. Have the authors made all data underlying the findings in their manuscript fully available?

Reviewer #1: Yes

Reviewer #2: Yes

5. Is the manuscript presented in an intelligible fashion and written in standard English?

Reviewer #1: Yes

Reviewer #2: Yes

6. Review Comments to the Author

Reviewer #1: The authors have exhaustively responded to previously requested minor revisions.

No further additions are required, the paper analyzes through a Delphi Survey the impact of Machine Learning methodologies in US Primary Care

Reviewer #2: Blease et al. addressed the minor point I raised and I am in favor of publishing this article in PLoS ONE.

7. PLOS authors have the option to publish the peer review history of their article (what does this mean?). If published, this will include your full peer review and any attached files.

Reviewer #1: **Yes: **Calogero Casà

Reviewer #2: No

---

## [Editor Report · Acceptance letter]

29 Sep 2020

PONE-D-20-12979R1 

US Primary Care in 2029: A Delphi Survey on the Impact of Machine Learning 

Dear Dr. Blease:

I'm pleased to inform you that your manuscript has been deemed suitable for publication in PLOS ONE. Congratulations! Your manuscript is now with our production department. 

Kind regards, 

on behalf of

Dr. Claudia Marotta 

Academic Editor

PLOS ONE